# Sanitary Emergencies at the Wild/Domestic Caprines Interface in Europe

**DOI:** 10.3390/ani9110922

**Published:** 2019-11-05

**Authors:** Luca Rossi, Paolo Tizzani, Luisa Rambozzi, Barbara Moroni, Pier Giuseppe Meneguz

**Affiliations:** Department of Veterinary Science, University of Turin, 10095 Grugliasco, Italy; paolo.tizzani@unito.it (P.T.); luisa.rambozzi@unito.it (L.R.); barbara.moroni@unito.it (B.M.); piergiuseppe.meneguz@unito.it (P.G.M.)

**Keywords:** transmissible diseases, livestock/wildlife interface, sylvatic reservoir, Europe

## Abstract

**Simple Summary:**

Even if it is an important achievement from a biodiversity conservation perspective, the documented increase in abundance of the four native European wild Caprinae (*Rupicapra rupicapra*, *R. pyrenaica*, *Capra ibex*, *C. pyrenaica*) can also be a matter of concern, since tighter and more frequent contact with sympatric livestock implies a greater risk of transmission of emerging and re-emerging pathogens. This article reviews the main transmissible diseases that, in a European scenario, are of greater significance from a conservation perspective. Epidemics causing major demographic downturns in wild Caprinae populations during recent decades were often triggered by pathogens transmitted at the livestock/wildlife interface.

**Abstract:**

Population density and distribution of the four native European wild Caprines (*Rupicapra rupicapra*, *Rupicapra pyrenaica*, *Capra ibex*, *Capra pyrenaica*) have increased in recent decades. The improved conservation status of this valuable wildlife, while a welcome event in general terms, is at the same time a matter of concern since, intuitively, frequent and tighter contacts with sympatric livestock imply a greater risk of cross-transmission of emerging and re-emerging pathogens, and offer unexpected opportunities for pathogens to spread, persist and evolve. This article recalls the transmissible diseases that are perceived in Europe to be of major significance from a conservation perspective, namely brucellosis (BRC) by *Brucella melitensis*, infectious kerato-conjunctivitis (IKC) by *Mycoplasma conjunctivae*, pestivirosis (PV) by the border disease virus strain 4 and mange by *Sarcoptes scabiei*. Special emphasis has been put on the epidemiological role played by small domestic ruminants, and on key knowledge needed to implement evidence-based prevention and control strategies. Remarkably, scientific evidence demonstrates that major demographic downturns in affected wild Caprinae populations in recent decades have often been triggered by pathogens cross-transmitted at the livestock/wildlife interface.

## 1. Introduction

The population density of the four native European wild Caprinae, namely, the Northern chamois *Rupicapra rupicapra*, Southern chamois *Rupicapra pyrenaica*, Alpine ibex *Capra ibex* and Iberian ibex *Capra pyrenaica*, has continued to increase in recent decades [1,2]. In parallel, the distribution area of these species has remarkably widened (Figure 1 and Figure 2). 

The natural colonization of low altitude mountains by the Northern chamois [3] and the reintroduction of the Alpine ibex, Abruzzo chamois *R. pyrenaica ornata* and Iberian ibex in mountain ranges where they became extinct during the 19–20th centuries [4,5,6] are well-known examples of this ongoing trend. Increased abundance of these mountain-dwelling ruminants is a welcome event that represents the prerequisite for: (1) the implementation of far-sighted management strategies of primary resources, such as alpine meadows and the montane mixed forests; (2) the conservation of key animal species, particularly large predators; (3) the strengthening of environmental awareness and culture amongst citizens, favoring a greater detectability of wild Caprinae compared to other wildlife; (4) the strengthening of wildlife-related tourism, spanning from hunting to targeted gastronomy, photo- and video documentation and guided naturalistic observations. However, frequent and tighter contacts between livestock and wild ruminants intrinsically imply a risk of the cross-transmission of emerging and re-emerging pathogens. Several diseases and parasites of wild Caprinae are shared with domestic ruminants due to close phylogenetic relationships, particularly sheep and goats. In most of the European mountain ranges, wild and domestic Caprinae are used to sharing the range on a seasonal basis (late spring and summer), when transhumant flocks are moved to sub-alpine and alpine meadows. During this period, domestic and wild species may get in direct or indirect contact [7], permitting the cross-transmission of pathogens [8]. The aim of this article is to recall infectious and parasitic diseases that are currently perceived of as major of concerns in native wild Caprinae in Europe. The term “concern” is intended here as “concern in a conservation perspective”, namely, because of the direct demographic impact of the diseases and the indirect effects due to possible wildlife population reduction, as a preventive and control measure, within national or continent-wide eradication programs in livestock. Special emphasis will be put on the epidemiological role played by sympatric wild and domestic Caprinae, and on the applicability of prevention and control strategies to reduce the impact of the concerned diseases. This review was conducted using a two-step approach. Firstly, diseases of major concern were identified based on presentation topics from the past 25 years at meetings of the European Wildlife Disease Association (EWDA) [9] and Groupe d’Etude sur l’Ecopathologie de la Faune Sauvage de Montagne (GEEFSM) [10], specifically, renewed spots of aggregation for specialists in the diseases of mountain-dwelling ruminants in Europe. Secondly, relevant research papers were searched and selected through three electronic databases (Scopus, PubMed, Web of Science) by taking into account coherence with the specific focus and ranking of the journals. Selected emergencies and related geographic distribution are summarized respectively in Table 1, Table 2 and Table 3.

## 2. Major Emergencies in Native Wild Caprinae in Europe

A first group of emergencies is represented by infections that are the objects of nationwide or continent-wide eradication programs, in particular brucellosis (BRC) by *Brucella abortus* and *B. melitensis*, and bluetongue by the Bluetongue virus. 

In the case of brucellosis, up until the end of the past century, it was a largely shared view in Europe that native wild ruminants (including Caprinae) were only playing a role of “dead-end” hosts. Outbreaks in wildlife were rare, self-limiting and traceable back to sympatric infected livestock (including cattle), while no spillback transmission to sympatric livestock was ever reported [11,12,13,14]. Interestingly, the timing of these outbreaks overlapped with the very last phases of the BRC eradication process in livestock in the respective countries. This consolidated view has been reconsidered due to the recent detection of a BRC outbreak by *B. melitensis* biovar 3 amongst Alpine ibex in the Bargy massif (Haute Savoie, French Alps). Somehow, the “Bargy crisis” has for the first time in decades put Europe in a sanitary emergency, which scientists and resource managers in North America are facing with a bison (*Bison bison*)/elk (*Cervus canadensis*)/cattle multi-host model [15]. As a recent example, 17 instances of BRC transmission from recognized wild reservoirs to livestock including elk, BRC-free cattle and ranched bison were detected between 2001 and 2012 in the Greater Yellowstone Area (Rocky Mountains, US), after an 11-year absence of BRC transmission [16].

In Bargy, the index case was established following raw milk cheese consumption by two children. A small single cattle herd out of 12,000 domestic ruminant heads raised in the area was found infected, while approximately half of the ibex that were chemically captured for active BRC surveillance (Alpine ibex is a protected species in France) tested positive for anti-*Brucella* antibodies [17]. Interestingly, (1) the dynamic circulation of *B. melitensis* biovar 3 amongst Bargy ibex was demonstrated a decade after BRC had been declared officially eradicated amongst local livestock in 1999; (2) the *Brucella* biovar and strain isolated in the children, cattle and ibex were the same as in the last outbreak farm before official BRC eradication [18]. As a result, BRC has been silently circulating in a sylvatic reservoir host for more than 10 years, showing that *Brucella*-susceptible wildlife should not be ignored in the design of *Brucella* eradication programs wherever a livestock/wildlife interface is present. The difficult management of the Bargy outbreak due to a mix of political and technical reasons (e.g., the effective application to free-ranging wildlife of vaccination and/or test-and-slaughter strategies, which are in common use in livestock) has been illustrated in scientific reports and informative articles [19,20,21].

Bluetongue (BT), a viral vector-borne disease, has re-emerged in Europe in the form of wide outbreaks amongst cattle and small domestic ruminants since the early 2000s, causing important economic losses [22]. Usually, while sheep are severely affected [23], wild Caprinae usually do not develop the clinical disease, nor do their populations suffer any decrease in number following spillover transmission from the domestic reservoir hosts [24]. Experimental infections and surveillance studies have demonstrated that red deer *Cervus elaphus* carry the BT virus asymptomatically for long periods, and may eventually play a maintenance host role, as suspected in zones where BT outbreaks have no longer been detected in livestock for spans of years [16,17,18]; as the opposite, wild Caprinae appear as dead-end hosts for BT, based on low viral titers and short periods of viremia. Low seroprevalence values and rare findings of BT RNA have been reported in *Rupicapra* and *Capra* spp. [24,25,26,27,28,29,30]. In addition to host-related risk factors, altitude was shown to limit the vector potential of *Culicoides* spp., hence the spread of BT [31,32].

A second group of emergencies has been seen in a limited number of disease outbreaks in domestic and wild ruminants, and has been characterized by a remarkable spreading potential over large areas of thousands of square kilometers. Four of these diseases have been renewed for their clear—sometimes catastrophic—impact on wild Caprinae populations: infectious kerato-conjunctivitis (IKC), pestivirosis (PV), sarcoptic mange (SM) and the “transmissible pneumonias” (TPs) complex. Controversial conclusions have been reached [33] in different wild ruminant models whether the causative agents of TPs act as primary or secondary pathogens or even commensal bacteria. Henceforth, since inconclusive results have been obtained so far in Europe on the debated etiology of TPs [34], little can be inferred on the cross-transmission of the candidate pathogens at the livestock/wildlife interface. Accordingly, this contribution will only deal with the remaining three diseases. 

Outbreaks of IKC in wild Caprinae, particularly *Rupicapra* spp. and *Capra* spp., are due to virulent strains of *Mycoplasma conjunctivae*, an atypical bacterium of the class Mollicutes that lacks a cell wall and hence has extremely poor off-host environmental persistence. Transmission occurs via direct contact and, eventually, via mechanical eye-to-eye transport operated by flies over short distances [35]. In the frame of a long-term study on a protected *R. pyrenaica* population, clear connections between the spread of IKC and the space use by different social units of chamois have been shown [36]. Onset of a novel IKC outbreak in unaffected wild Caprinae is usually perceived as a “summer crisis”, characterized by high incidence and relatively low mortality and fatality rates. In addition, the sex- and age-biased distribution of clinical cases is more often detectable in females than in males, and in adult females compared with kids. The symptoms and behavior of affected chamois and the presence of orphan kids have a strong emotional impact on mountain visitors, who are especially numerous and active in the season. In contrast, the spread of IKC during winter is characterized by lower incidence and higher mortality and fatality rates, with lower visibility by a large public. Overall, mortality rates are usually low, but can reach up to 30% due to blindness-related consequences such as falls or drowning by affected animals [35,37,38]. A striking characteristic of IKC outbreaks is the potential to rapidly spread over large areas, at a speed of over 15 km/year from the index case [39], due to efficient direct and indirect transmission [40,41]. Since the final decades of the past century, major IKC outbreaks have been reported in the Alps, Pyrenees and Cantabrian Mountains. In recent years, outbreaks with a particularly wide extension—the largest ever known—have occurred in the Western Alps and in the Central Pyrenees [40,42]. However, sporadic cases and small foci with limited spatial spread are also known to wildlife professionals [42]. Apparently, IKC does not persist endemically in the same area and, at a small spatial scale (up to few thousand hectares), the epidemic wave usually vanishes within a few months, suggesting an important role by herd immunity. *M. conjunctivae* is frequently isolated from the eyes of sheep and goats, and IKC is a well-known condition in small ruminant farms worldwide. Sheep to wild Caprinae transmission has been experimentally reproduced [35]. Until recently in Europe it was largely accepted that: (1) domestic flocks are the main reservoir of *M. conjunctivae*; (2) IKC outbreaks in wild Caprinae originate, most frequently, from occasional spillover of virulent strains from domestic reservoirs; (3) *M. conjunctivae* infection does not persist autonomously in chamois or any other wild Caprinae populations [29]. Amongst other evidence, a strong argument in favor of this “classic” view was the rare and timely spaced occurrence of IKC outbreaks in chamois herds subjected to long-term monitoring within National Parks and Hunting Reserves in Europe. Furthermore, the first occurrence of epidemic IKC in chamois in New Zealand occurred 40 years after the introduction of eight chamois from the Austrian Alps. In recent years, the aforementioned “classic” view has been put into question by the molecular-based isolation of *M. conjunctivae* in the eyes of healthy chamois in Switzerland [30]. Although it is not clear if positive asymptomatic animals are truly healthy carriers or individuals in incubation phase not yet cleared of a previous infection, such findings suggest that “an endemic presence of *M. conjunctivae* in wild mountain ungulates cannot be excluded on large territories used by interconnected subgroups of wild ungulates, although this would not rule out sheep as a potential source of infection” [43,44]. Interestingly, and similar to the case of brucellosis, Alpine ibex seem to be a better candidate than chamois in playing a reservoir role complementary to domestic flocks [45]; nevertheless, epidemiological studies are still needed to define if sympatry with ibex is a risk factor for occurrence of IKC outbreaks in chamois. Most recently, persistence of *M. conjunctivae* strains in wild Caprinae in the Alps was robustly suggested by merging field observations and molecular analyses [28]. Moreover, independent *M. conjunctivae* sylvatic and domestic cycles were shown to coexist in the Pyrenees, with sheep and chamois (*R. pyrenaica*) as key host species [46]. It follows that if the cross-transmission of the agent between domestic and wild Caprinae is a relatively rare event [46], little or no benefit to unaffected wildlife herds can be expected from implementing measures such as the enhanced clinical surveillance of transhumant flocks or the isolation and treatment of clinically affected individuals prior to being moved uphill. 

In the early 2000s, a novel Pestivirus fam. Flaviviridae of the “border disease virus” (BDV) group was isolated in sick chamois in the Central and Eastern Pyrenees in Spain and France [47,48]. The agent of pestivirosis (PV) in chamois is described as a specific variant belonging to the BDV4 genotype, which the same genotype as the BDV is circulating in sheep in Spain [49]. PV in chamois is clinically characterized by variable degrees of cachexia, an alopecia often associated with skin hyperpigmentation and neurological disorders such as depression, weakness and difficulty in moving prior to death. Signs of secondary infections magnified by the immunosuppressive effects of BDV4 infection (e.g., dyspnea due to bacterial bronchopneumonia) have also been a frequent finding [50]. The demographic effects of PV are extremely variable, from a mild impact on reproductive performance to severe die-offs with mortality rates between 40% and 85%, as observed in the Eastern Pyrenees [49]. The reasons for such variability, still to be fully elucidated, include the viral strains involved, the epidemiological phase of the infection (epidemic versus endemic), the herd immunity eventually influenced by contacts with related viral strains of domestic origin and the social and spatial structure of the affected host populations and their genetic variability [51,52]. Nevertheless, it is estimated that the whole chamois population of the Central and Western Pyrenees has decreased in number by approximately one-third since 2001 [50]. Persistence of BDV4 infection after a first PV outbreak has been demonstrated, as was the opposite scenario of viral extinction. Recovery of the affected chamois populations was weak in the first case and rapid in the second. However, in the latter population, the return to a viral and serological naïve status is now a matter of concern for resource managers, since BDV4-infected chamois are still present in neighboring chamois herds [51]. As to the origin of this emerging conservation medicine problem, an innovative phylogenetic study of available viral sequences suggests that: (1) the chamois clade originated from sheep BDV4, generating a founder effect; and (2) the “capture” by the new sylvatic host was a recent event, datable back to approximately two decades ago. In addition, the study shows that intra-specific subclading of the border disease (BD) “chamois” variant is already detectable along the Pyrenees [53]. Nowadays, sheep and goats have apparently no role in maintaining the circulation of the “chamois strain” of BD in the Pyrenees. Accordingly, prophylactic control of BD in livestock whenever feasible (an effective and safe vaccine is currently not available) would be of limited interest for PV management from a conservation perspective. As an alternative, it is tempting to figure out if contact with domestic flocks endemically infected by “their” BD strains would result in a sort of natural and beneficial cross-vaccination of sympatric chamois, possibly enhancing herd immunity against the “chamois strain” or other putative novel “chamois” strains in future. If this were the case (though field and experimental studies are yet necessary to confirm it), sympatry with such flocks would be desirable, in contrast with the traditional view by local resource managers. Similarly, implementation of active sero-surveillance schemes in managed chamois populations is warranted to check the herd immunity in front of any variants of the BDV, as well as the design management options according to the epidemiological status, without inappropriate generalizations [54]. Outside the Pyrenees, seroreactors to pestiviruses with an origin in livestock have been frequently found amongst chamois surveyed in the Western Alps [55,56] and, to a lesser extent, in the Central Alps and Cantabrian Mountains [57,58]. In contrast, no seroreactors were found in an isolated endangered chamois population (*R. pyrenaica ornata*) in the Apennines (Italy) [59], and they are rare amongst Alpine and Spanish ibex [29,59]. 

Sarcoptic mange (SM) is caused by the burrowing mite *Sarcoptes scabiei*. Several varieties of the mite have been traditionally described as able to successfully infect a limited range of zoologically related hosts. For example, mites infecting chamois in the Alps, usually referred as *S. scabiei* var. *rupicaprae*, are naturally or experimentally cross-transmissible to the Alpine ibex and the domestic goat, as well as, less frequently, to domestic sheep, mouflon *Ovis aries musimon*, roe deer *Capreolus capreolus* and red deer *C. elaphus* [60]. As of 1987, there was a single wide SM outbreak area affecting chamois in the Eastern Alps across Austria, Germany, Slovenija and Italy eastbound along a line connecting two large rivers, the Inn and Adige [61]. Since then, new outbreak areas have been reported in several mountain systems in Southern and Eastern Spain, affecting the *hispanica* subspecies of *C. pyrenaica* [62,63,64], the sympatric exotic *Ammotragus lervia* [65] and the Eastern population of *R. pyrenaica parva* [64] in the Cantabrian Mountains. Currently, the Southern chamois in the Pyrenees and Apennines, the Northern chamois and Alpine ibex in the Western and Central Alps and the *victoriae* subspecies of the Iberian ibex enjoy an SM-free status. Persistence for centuries (as in the case of the Austrian Alps) and the relatively low spread of 3–6 km/year on average are well known characteristics of SM in *Rupicapra* spp., while several thousands of hectares/year are rapidly infected when representatives of the genus *Capra* are involved [62]. Mortality by SM typically peaks in winter and spring, and it is influenced by interaction with natural factors such as winter starvation and other climate constraints [66]. On a larger temporal scale, mortality is mainly related to the life history of affected populations, namely to the previous contacts (or not) with the agent. In the case of a first epidemic wave of SM in previously unaffected areas, the demographic impact may be remarkable. In the recently affected Dolomite Alps and Cantabrian Mountains, chamois population size has decreased on average by approximately two-thirds [66,67], and up to more than 80%. A 98% decrease rate occurred in a naïve and particularly sensitive population of Spanish ibex in Southern Spain [68]. Alternatively, in the case of successive contacts (usually occurring in form of minor waves at 10–15 years intervals), mortality rarely exceeds a value 25% [69]. In the Cantabrian Mountains, a new equilibrium characterized by a population size of approximately two-thirds of the pre-outbreak population was reached in the last decade, as the effect of the first epidemic wave and the subsequent endemic phase of the disease [67]. Other factors may influence the short and long-term outcome of SM, including the host’s genetic structure and variability [70,71]. Under field conditions, the responsibility of infected domestic goats (and, less likely, sheep) in triggering SM into naïve wild Caprinae populations has been suspected, though not unambiguously demonstrated [63,65,70,71,72]. Experimental infection trials have been nonetheless successfully carried out in both directions [72], and a spontaneous SM spillback episode in domestic goats, originating from contacts with naturally infected chamois, has been reported [73]. Finally, chamois are more likely to infect other sympatric wildlife than the opposite, as occurred in the case of the several colonies of Alpine ibex in the Eastern Alps [74], and of a single isolated colony of Iberian ibex *C. pyrenaica* in the Cantabrian Mountains. From a conservation perspective, attention to the livestock/wildlife interface should be focused on preventing infected goat flocks from being introduced into areas that are home to SM-free populations of wild Caprinae, such as the Western and Central Alps, Apennines, Spanish Central System and Carpathians. Awareness of this neglected trade-related risk should be raised at international and national institutional levels. 

## 3. Conclusions

In this review, evidence or sound plausibility has been provided that some of the major outbreaks reported in wild Caprinae in Europe were triggered by pathogens initially cross-transmitted at the interface with livestock (primarily sheep and goats). When the focus is shifted from triggering to persistence in wildlife populations, trials spanning over several decades have unambiguously shown that:(1)sheep and goats can no longer be considered the exclusive or the main reservoir of *M. conjunctivae* for wild Caprinae;(2)following suspected infection from livestock, chamois (*Rupicapra* spp.) and ibex (*Capra* spp.) have rapidly turned into the exclusive reservoir hosts of *S. scabiei* wherever SM outbreaks have been reported in wild ruminant hosts; and(3)similar to (2), Southern chamois *R. pyrenaica pyrenaica* is currently deemed to be the single reservoir of the specific BDV4 variant, and the Alpine ibex is considered the single reservoir of *B. melitensis* in the French Alps.

Little can be done at the livestock/wild Caprinae interface to control emergencies. As an exception (the single one to our knowledge), cohabitation of chamois with BDV-infected sheep should be encouraged in order to strengthen herd immunity against the specific BDV-4 strain [49]. In general, prevention of BDV outbreaks seems unlikely and, due to current limitations (including the unavailability of performant vaccines [74]), it would be difficult to prevent new BDV livestock strains from adapting to chamois, as occurred in the Central Pyrenees. In contrast, the active surveillance of transhumant goat flocks before moving uphill in cohabitation with wild Caprinae, and their mass treatment with effective acaricides, seem pivotal measures to reducing the risk that SM may spread from scabietic or carrier domestic goats to naïve herds of their sylvatic relatives. Europe-wide eradication of SM in goats should be the optimal goal to achieve in the near future. As of BRC in the French Alps, the “problem” is the possible spillback to sanitated livestock. Evidence shows that this risk has remained extremely low for more than 25 years, when *B. melitensis* was established as a new unexpected wild reservoir host. Awareness of the limited spillback risk in the Bargy massif does not impede improved biosecurity measures, including fencing and the use of shepherd dogs, that can be adopted at the livestock/wildlife interface, or that the adaptive management approach currently in progress (e.g., capture followed by test-and-slaughter or test-mark-and release according to results of a rapid BRC diagnostic test) be prolonged in time to reduce the prevalence of BRC in these particular ibex [20]. In conclusion, it is our opinion that in mountain systems in Europe, the conservation-oriented management of animal health at the livestock/wildlife interface should prioritize measures and strategies aimed to prevent the introduction of a limited range of “major” pathogens into naïve herds of wild Caprinae. To achieve this ambitious goal, the limited resources should target applied research and the active surveillance of these pathogens in view of an early detection in putative domestic sources, with a special focus on uphill transhumant flocks.

## Figures and Tables

**Figure 1 animals-09-00922-f001:**
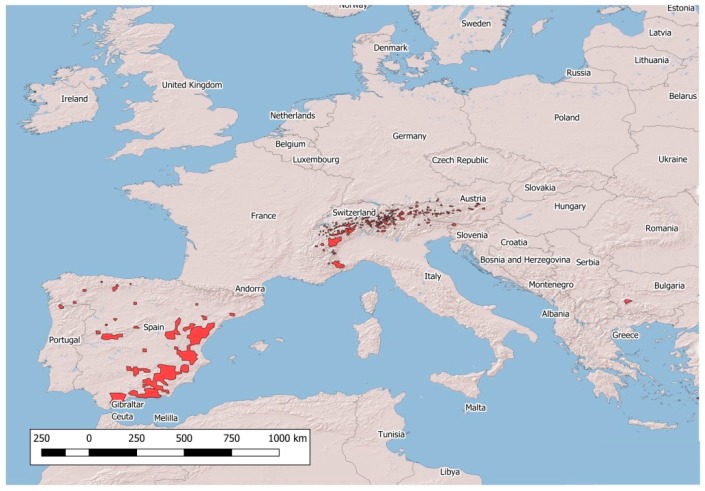
Distribution of native wild *Capra* spp. in Europe.

**Figure 2 animals-09-00922-f002:**
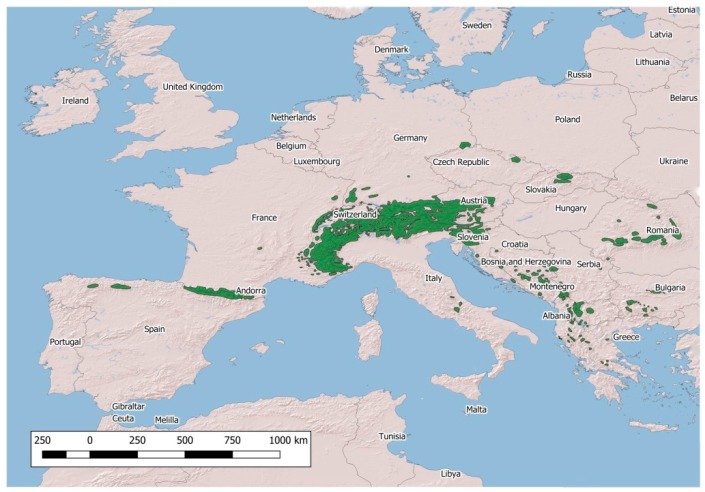
Distribution of native *Rupicapra* spp. in Europe.

**Table 1 animals-09-00922-t001:** Major transmissible diseases at the livestock/wild Caprinae interface in Europe.

Name of the Disease	Pathogen	Wild Hosts	Eradication Program in Livestock
Blue tongue	Blue tongue virus (Reoviridae)	Northern Chamois Southern Chamois Alpine Ibex Iberian ibex	yes
Pestivirosis	Border disease virus (strain 4)	Southern Chamois	no
Brucellosis	*Brucella melitensis*	Northern Chamois Alpine Ibex Iberian ibex	yes
Infectious kerato-conjunctivitis	*Mycoplasma conjunctivae*	Northern Chamois Southern Chamois Alpine Ibex Iberian ibex	no
Sarcoptic mange	*Sarcoptes scabiei*	Northern Chamois Southern Chamois Alpine Ibex Iberian Ibex	no
	*(1)*		

**Table 2 animals-09-00922-t002:** Reported outbreaks of selected diseases in *Capra* spp. in Europe.

Mountain System	*Capra* spp.				
	BRC	BT	IKC	PV	SM
Penibaetic System	+	+	+	-	+
Central System	-	-	-	-	-
Iberian System	-	-	-	-	+
Pyrenees (E)	-	-	-	-	-
Pyrenees (F)	-	-	+	-	-
Western Alps (F)	+	-	+	-	-
Western Alps (I)	+	-	+	-	-
Western Alps (CH)	-	-	+	-	-
Eastern Alps (I)	-	-	-	-	+
Eastern Alps (A, D)	-	-	+	-	+
Eastern Alps (SLO)	-	-	-	-	+

BRC: brucellosis; BT: bluetongue; IKC: infectious kerato-conjunctivitis; PV: pestivirosis; SM: sarcoptic mange.

**Table 3 animals-09-00922-t003:** Reported outbreaks of selected diseases in *Rupicapra* spp. in Europe.

Mountain System	*Rupicapra* spp.				
	BRC	BT	IKC	PV	SM
Cantabrian Mountains	-	-	+	-	+
Pyrenees (E)	-	-	+	+	-
Pyrenees (F)	-	-	+	+	-
Western Alps (F)	+	-	+	-	-
Western Alps (I)	+	-	+	-	-
Western Alps (CH)	-	-	+	-	-
Eastern Alps (I)	-	-	+	-	+
Eastern Alps (A, D)	-	-	+	-	+
Eastern Alps (SLO)	-	-	+	-	+
Tatra Mountains	-	-	-	-	-
Carpathian Mountains	-	-	-	-	-
Balkan Mountains	-	-	-	-	-

BRC: brucellosis; BT: bluetongue; IKC: infectious kerato-conjunctivitis; PV: pestivirosis; SM: sarcoptic mange.

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
