# Peer review of "Sanitary Emergencies at the Wild/Domestic Caprines Interface in Europe"

_animals, 2019, doi:10.3390/ani9110922_

Round 1
Reviewer 1 Report
Please see attached file.

Author Response
To the attention of the Reviewer.
We are pleased to receive your constructive feedback and the excellent comments from the Reviewers. The manuscript has been carefully revised following the major comments and guidance addressed.
All the points that have been raised by the reviewers were thoroughly considered and addressed, and we provide a detailed response to each of them in the attached file.
We thank you for your time and consideration, and we look forward to hearing from you.
Yours sincerely,
Luisa Rambozzi, on behalf of all authors
luisa.rambozzi@unito.it

Reviewer 2 Report
The subject of the article is very interesting but on my mind, it is possible to improve some points.
Major points:
The aim of the article is unclear. The authors in line 58 wrote: “The aim of this article is recalling which, amongst these pathogens and related pathologies, are currently perceived of major concern in the European scenario. Which scenario? Major concern is defined as threat for wildlife or domestic ruminants?
The article focused on several diseases but authors did not justify their choice. There is no methodology of the review (keywords in Scopus, Pubmed…) and therefore, why did you choose SM whereas you said in line 231: “livestock does not play any significant role in the dynamics of this disease in the latter”
In the conclusion, authors said in line 247: “Little can be done at the livestock/wild Caprinae interface to counteract running emergencies.” Why you did not discuss about biosecurity measures to segregate wild from domestic ruminants.
Minor points :
Line 87: in case of brucellosis, the trouble with vaccination is more the availability of effective and secure vaccine than his effective application
Lines 64 and 98: the first group is represented by infections which are the object of nation-wide eradication programs and the second group is represented by infections which are not the object of nation-wide eradication programs but characterized by remarkable spreading potential over large areas. That’s right ? The distinctions between these two groups of infections are not clear may be in link with the lake of explanation of methodology.
Line 129: there is no explanation of effects of infection in domestic animals
Author Response
To the attention of the Reviewer.
We are pleased to receive your constructive feedback and the excellent comments from the Reviewers. The manuscript has been carefully revised following the major comments and guidance addressed.
All the points that have been raised by the reviewers were thoroughly considered and addressed, and we provide a detailed response to each of them in the attached file.
We thank you for your time and consideration.
Yours sincerely,
Luisa Rambozzi, on behalf of all authors

Round 2
Reviewer 2 Report
Thank you for taking into account my comments.